# Genetic Control of MAP3K1 in Eye Development and Sex Differentiation

**DOI:** 10.3390/cells11010034

**Published:** 2021-12-23

**Authors:** Jingjing Wang, Eiki Kimura, Maureen Mongan, Ying Xia

**Affiliations:** Department of Environmental and Public Health Sciences, College of Medicine, University of Cincinnati, Cincinnati, OH 45267-0056, USA; wangjingjing_wjj@126.com (J.W.); kimuraei@ucmail.uc.edu (E.K.); monganmc@UCMAIL.UC.EDU (M.M.)

**Keywords:** MAP3K1, JNK, dioxin, genetic crosstalks, gene-environment interactions, embryonic eyelid closure, sex development and differentiation

## Abstract

The MAP3K1 is responsible for transmitting signals to activate specific MAP2K-MAPK cascades. Following the initial biochemical characterization, genetic mouse models have taken center stage to elucidate how MAP3K1 regulates biological functions. To that end, mice were generated with the ablation of the entire *Map3k1* gene, the kinase domain coding sequences, or ubiquitin ligase domain mutations. Analyses of the mutants identify diverse roles that MAP3K1 plays in embryonic survival, maturation of T/B cells, and development of sensory organs, including eye and ear. Specifically in eye development, *Map3k1* loss-of-function was found to be autosomal recessive for congenital eye abnormalities, but became autosomal dominant in combination with *Jnk* and *RhoA* mutations. Additionally, *Map3k1* mutation increased eye defects with an exposure to environmental agents such as dioxin. Data from eye developmental models reveal the nexus role of MAP3K1 in integrating genetic and environmental signals to control developmental activities. Here, we focus the discussions on recent advances in understanding the signaling mechanisms of MAP3K1 in eye development in mice and in sex differentiation from human genomics findings. The research works featured here lead to a deeper understanding of the in vivo signaling network, the mechanisms of gene–environment interactions, and the relevance of this multifaceted protein kinase in disease etiology and pathogenesis.

## 1. Introduction

The mitogen-activated protein kinases (MAPKs) play pivotal roles in diverse cellular activities such as gene expression, cell proliferation, migration, survival, and death. In eukaryotes, there are three major MAPK subgroups: the extracellular signal-regulated kinases (ERKs), the Jun-N-terminal kinases (JNK), and the p38s [1]. Each subgroup is controlled by specific MAP kinase kinases (MAP2Ks) that phosphorylate and activate the MAPKs. Generally speaking, the MAP2K1/2 are upstream kinases of the ERK1/2, the MAP2K4/7 are upstream activators of the JNK1/2, and the MAP2K3/4/6 are responsible for the activation of the p38s. In turn, the MAP2Ks are activated through phosphorylation by the MAP kinase kinase kinases (MAP3Ks). The MAP3K is a large superfamily consisting of at least 19 protein kinases [2]. Members of this family share relatively similar kinase domains but rather distinct regulatory regions. The regulatory domains mediate interactions with regulators, adaptors, and structural components to crucially determine the specificity and compartmentalization of signal transduction. Collectively, the different MAP3Ks enable the transduction of diverse signals to specifically activate the MAPK pathways.

MAP3K1, also known as MEK Kinase 1 (MEKK1), was identified over twenty years ago as a member of the MAP3K family [2,3]. Biochemistry data show that MAP3K1 preferentially activates MAP2K4 and MAP2K7, which in turn activate the JNKs and/or p38 MAPKs. While MAP3K1 is widely considered an upstream enzyme of the JNK/p38 pathways, it is occasionally capable of regulating the ERK and nuclear factor-κB (NF-κB) pathways [4].

As a large protein of around 1500 amino acids, MAP3K1 possesses distinct structural domains that bring about various regulatory and functional activities (Figure 1). The kinase domain (KD), located at the C-terminus, is responsible for the interaction with and the phosphorylation of the downstream MAP2Ks [5]. The DEVD sequences, upstream of the kinase domain, are targets of caspase-3 cleavage. Caspase-3 activation by genotoxic agents [6], ischemia-injury [7], and TNFα [8] causes MAP3K1 proteolytic cleavage, resulting in a *circa* 90 kDa C-terminal fragment that promotes cell apoptosis. The long region N-terminal of the KD has a number of well-defined functional domains, including SWI2/SNF2 and MuDR (SWIM), RING finger (RING), Armadillo Repeats (ARM), and Tumor Overexpressed Gene (TOG). Of these, the SWIM domain mediates protein–protein interactions and binds to c-Jun to facilitate the subsequent c-Jun ubiquitination and degradation [9]. The RING domain has a typical Plant Homeodomain (PHD) motif, closely related to a RING finger with the specific spacing of cysteine and histidine residues. This domain has an E3 ubiquitin ligase activity that promotes the ubiquitination/degradation of a number proteins, including ERK, c-Jun, TTP, TAB1, and CARMA1 [10,11,12,13,14]. MAP3K1-mediated protein ubiquitination regulates cell survival and death in response to cytokine and stress signals [10,15,16]. At the N-terminus, MAP3K1 has a putative Guanine Exchange Factor (GEF) domain that mediates interactions with the small GTPases RAC and RHOA; it also binds with α-actinin through residuals 221–559. These interactions are believed to enable MAP3K1 to modulate the cytoskeleton [17,18,19]. Recently, a TOG domain has been identified in the region overlapping with the ARM that mediates interactions with AXIN1 [20,21]. The TOG domain preferentially binds with curved tubulin heterodimers, notwithstanding the biological functions of the MAP3K1–tubulin interactions are unclear [17,22].

In vitro studies have also revealed diverse putative signals and potential mechanisms for the activation of MAP3K1. For instance, the expression of the dominant negative MAP3K1 blocks the activation of JNK and/or ERK by inflammatory cytokines such as TNFα and INFγ [14,23]. The cytokine signals are likely mediated through the activation of the MAP4Ks, including the Germinal Center Kinase (GCK) [24], the Hematopoietic Progenitor Kinase 1 (HPK1) [25], and the Nck Interacting Kinase (NIK) [26], which bind directly to and phosphorylate MAP3K1. Additionally, MAP3K1 oligomerization induces autophosphorylation at T1381 and T1393, leading tothe kinase activation [24,27], whereas oxidative stress causes the glutathionylation of MAP3K1 at C1238 to inhibit the kinase activity [28]. In another instance, cells with genetic MAP3K1 ablation are defective in the induction of MAPK activities and cell migration by the growth factors TGFβ and EGF [29,30]. The EGF induces adaptor protein GRB2–MAP3K1 interactions to activate the JNK pathway [31], but hypoxia activates MAP3K1 to induce ERK signaling [32].

Taken together, MAP3K1 is a protein with multiple functional domains and activated by a wide range of stimuli and mechanisms. The diverse ways of MAP3K1 regulation underscore its functional complexity. In this review, we will summarize the current understanding of the physiological roles of MAP3K1, with emphases on two emerging topics: (1) eyelid development based on genetic studies in mice and (2) sex development and differentiation based on human genomics research.

## 2. Diverse Roles of MAP3K1 Revealed in Genetic Mouse Models

MAP3K1 is highly conserved in mammals. The human and mouse orthologs share 88% sequence identity, suggesting a functional similarity between mice and men. In mouse development, MAP3K1 is detected as early as the 2-cell stage and becomes highly expressed in the brain, the glands, the metanephros, the sensory organs, and the skin by embryonic day (E) 15 [33,34]. MAP3K1 expression is also detectable, albeit at lower levels, in the embryonic heart, liver, ovary, and testis. The expression profile is consistent with the diverse roles MAP3K1 displays in organogenesis and tissue formation. Mice lacking the entire *Map3k1* gene [35], the coding sequences for the kinase domain [29], or carrying point mutations in the RING domain [12], display abnormalities in their vision/eye, the nervous system, hearing/vestibular/ear, immunity, and the cardiovascular system (Available online: http://www.informatics.jax.org/marker/MGI:1346872 (accessed on 1 December 2021)).

### 2.1. Full-Length MAP3K1 Ablation (Map3k1^−/−^) Mice

The *Map3k*^−/−^ mice carry genomic DNA deletions that remove 132 codons, including the ATG site in exon 1, resulting in the deletion of the entire MAP3K1 protein [36]. Although they survived embryonic development, the *Map3k1*^−/−^ mice were born with an eye open at birth (EOB) phenotype. In addition, these mice were susceptible to heart failure and sudden death following cardiac pressure overload [37], and also had delayed tumor cell dissemination and metastasis [38].

### 2.2. The Kinase Domain-Deficient (Map3k1^ΔKD^) Mice

Mice lacking the kinase domain of MAP3K1 (*Map3k1*^Δ*KD*^) are the most widely investigated in different research laboratories. In the *Map3k1*^Δ*KD*^ allele, the sequences coding for the MAP3K1 kinase domain are replaced with the β-galactosidase gene [39]. The kinase domain-deficient *Map3k1*^Δ*KD/*Δ*KD*^ mice were born with an EOB phenotype, the same as that seen in the *Map3k1*^−/−^ mice. In addition, these mice had compromised immunity, with reduced T-cell survival, aberrant differentiation of the intra-thymic CD4^+^ and CD8^+^ subsets, and deficient B-cell proliferation and antibody production in response to antigen [40,41,42]. Some of the immunological functions that depend on the autonomous MAP3K1 activity, because conditional deletion of *Map3k1,* specifically in T cells, abolished the invariant NK T cell proliferative expansion in response to glycolipid antigen [13]. Furthermore, although MAP3K1 is overall considered dispensable for embryonic survival, a considerable number of the *Map3k1*^Δ*KD/*Δ*KD*^ embryos died due to abnormalities in erythropoiesis in the C57BL/6 background, suggesting that MAP3K1 contributes to embryonic survival in a genetic background-dependent manner [43].

The *Map3k1*^Δ*KD*^ allele expresses a kinase domain defective MAP3K1 (N-terminal)-β-Gal fusion protein, controlled by the endogenous *Map3k1* promoter, and is detectable in situ by X-gal staining [29]. Using whole-mount X-gal staining, MAP3K1 expression was detected in the ear, specifically in the inner and outer hair cells, Claudius cells, Hensen cells, border cells of the internal spiral sulcus, and spinal ganglion neurons as well as the apical surface of supporting cells of the organ of Corti [44,45]. The *Map3k1*^Δ*KD/*Δ*KD*^ mice, accordingly, became severely deaf with the degeneration of outer hair cells in the organ of Corti at postnatal day (P)14 and the loss of cochlear spinal ganglions at P90 [45]. The organ of Corti in these mice had reduced the expression of FGF/FGFR genes important for otic placode induction and epithelium development [46,47]. It is worth noting that in an *N*-ethyl-*N*-nitrosourea (ENU) mutagenesis screen, Parker et al. identified a recessive mutant mouse, termed *goya*, carrying a mutation in the *Map3k1* gene [44]. The mutation was a single-nucleotide change in the splice donor site of intron 13 that resulted in a truncated MAP3K1 containing only the N-terminal half of the protein. The goya mice were therefore similar to the *Map3k1*^Δ*KD/*Δ*KD*^ mice in the sense that they expressed a kinase domain-deficient MAP3K1. Unsurprisingly, the goya mice phenocopied the *Map3k1*^Δ*KD/*Δ*KD*^ mice and displayed progressive hearing loss in addition to an EOB phenotype. The genetic data suggest that the MAP3K1 kinase domain is crucially required for ear/hearing and eye development.

### 2.3. The Ubiquitin Ligase Domain-Deficient Mice

A knock-in mouse strain that carries two mutations, C438A and I440A, in the MAP3K1 RING domain was generated. The mutations resulted in the loss of E3 Ub ligase activity [12]. The *Map3k1^mPHD/mPHD^* mice died in early embryogenesis, but the *Map3k1^mPHD/+^* mice survived development and exhibited enlarged testes and hearts, aberrant B-cell development, and T-cell signaling [12].

Considered together, the different mutant strains reveal a combination of the multifaceted physiological roles of MAP3K1 and its functional domains in the living organisms. Distinct from the ubiquitin ligase-deficient mice, which are embryonic lethal, the *Map3k1*^−/−^, *Map3k1*^Δ*KD/*Δ*KD*^, and goya mice survived embryogenesis and displayed a common birth defect of the eye. The next section details work that investigated the mechanisms of MAP3K1 in eye development.

## 3. The Roles of MAP3K1 in Embryonic Eye Development

### 3.1. The Signaling Mechanisms of MAP3K1 in Embryonic Eyelid Closure

Mouse eyelid development begins at embryonic day 11.5 (E11.5). At this stage, the periocular epithelium derived from the surface ectoderm folds at the junction of the future conjunctiva and cornea to form the initial eyelids. As the embryo grows, the eyelids continue to elongate, and the epithelium at the eyelid tip moves centripetally, leading ultimately to the fusion of the upper and lower eyelids [48], which in mice occurs between E15.5 and E16.5 [48,49]. The eyelids remain fused at birth and separate around postnatal day 12 (P12) as the result of apoptosis and the keratinization of epithelial cells at the fusion junction [50]. Therefore, mice are normally born with their eyes closed, whereas those with a failure of eyelid closure in embryogenesis exhibit an EOB phenotype [51] (Figure 2A).

As mentioned earlier, the MAP3K1-deficient strains, including *Map3k1*^−/−^, *Map3k1*^Δ*KD/*Δ*KD*^, and goya, which were generated independently in different laboratories, all displayed the EOB defects [52,53]. In addition, a mouse strain carrying a spontaneous 27.5 kb germline deletion that removes exons 2–9 and causes the frame-shift of the MAP3K1 protein (*Map3k1^lg-Ga^*) also had this phenotype [54]. The genetic evidence is compelling in support of the essential role of MAP3K1 in eye development.

A histological examination of prenatal eyes identified the developmental origin of the eye defects. At E15.5, prior to the onset of eyelid closure, the upper and lower eyelids were separated in both wild-type and *Map3k1*^Δ*KD/*Δ*KD*^ embryos; however, at E16.5 post eyelid closure, while the eyelids were clearly fused in wild-type fetuses, they were still manifestly separated in *Map3k1*^Δ*KD/*Δ*KD*^ fetuses [29] (Figure 2B). The *Map3k1*^Δ*KD/*Δ*KD*^ mice, therefore, have defective eyelid closure in embryonic development.

The X-gal staining of the *Map3k1*^Δ*KD*^ embryos detected MAP3K1 expression in various ocular tissues, including the lens, eyelid, and ciliary epithelium, as well as in the retinal progenitors and pigment epithelial cells [55] (Figure 2C). Consistent with the expression in the retina and the eyelids, the *Map3k1*^Δ*KD/*Δ*KD*^ mice had retina defects postnatally due to aberrant neuronal cell proliferation and apoptosis, besides the EOB phenotype [29,55]. In the embryonic eyelids, MAP3K1 was expressed abundantly in the inferior periderm near the leading edge, where the phosphorylation of MAP2K4, JNK, and c-Jun, a JNK downstream target, was elevated in wild-type but not *Map3k1*^Δ*KD/*Δ*KD*^ embryos [29,56] (Figure 2C,D). These observations suggest that MAP3K1 regulates a temporal–spatial activation/phosphorylation of JNK and c-Jun in the embryonic eyelid epithelium.

c-Jun is a member of the AP-1 family of transcription factors [57]. Conditional *Jun* knockout in keratinocytes (c-Jun-skin-null) produces an EOB phenotype, similar to that seen in the MAP3K1-deficient mice [58,59]. These observations were originally thought to be a resounding indication of a MAP3K1-c-Jun pathway in eyelid development. However, compelling evidence argues against this idea. The *Map3k1*^Δ*KD/*Δ*KD*^ eyelids decreased c-Jun phosphorylation but unaltered c-Jun expression [29]. Lacking c-Jun phosphorylation, however, is insufficient to block eyelid closure because mice harboring a phosphorylation-deficient c-Jun (*Jun^AA^*) have normal eyelid development [60]. In addition, MAP3K1 expression and c-Jun induction appear to be temporal-spatially segregated in the embryonic eyelids; MAP3K1 expression is abundant in the inferior periderm, while c-Jun induction is predominant in the epithelium at the leading edge [61]. Moreover, *Map3k1* and *c-Jun* do not display non-allelic non-complementation in genetic crossing experiments [61], strengthening the notion that MAP3K1 regulates eyelid development independent of c-Jun.

Searching for other downstream effectors of MAP3K1 led to the identification of abnormal cell–cell contacts and reduced actin polymerization in the eyelid epithelium of the *Map3k1*^Δ*KD/*Δ*KD*^ embryos. In vitro, the *Map3k1*^Δ*KD/*Δ*KD*^ keratinocytes displayed defective F-actin formation and migration in comparison to their wild-type counterparts [29]. The MAP3K1-regulated cytoskeleton reorganization and epithelial cell migration therefore likely contribute to eyelid closure.

Recently, Chen et al. reported a new EOB strain derived from N-ethyl-N-17 nitrosourea mutagenesis, in which they detected a T941A mutation in exon 4 of *Map3k1*, resulting in an L314Q substitution [62]. In contrast to the other genetic strains where the *Map3k1* mutant allele is recessive, the *Map3k1^L314Q^* is a dominant pathogenetic allele. The *Map3k1^L314Q/+^* heterozygous mice exhibit EOB that is not observed in the heterozygous *Map3k1*^+/−^ and *Map3k1^+/^*^Δ*KD*^ mice. How MAP3K1(L314Q) induces eyelid closure defects is still a mystery. Since the affected amino acid is located near the SWIM domain that mediates c-Jun degradation, it is reasonable to speculate that the *Map3k1^L314Q/+^* mutation promotes c-Jun protein degradation, and thereby the EOB phenotype [9]. In support of the speculation, the *Map3k1^L314Q^* eyelids have indeed reduced c-Jun protein, along with decreased c-Jun phosphorylation [62].

### 3.2. Genetic Identification of the MAP3K1 Pathway in Eyelid Morphogenesis

Although the *Map3k1*^Δ*KD/*Δ*KD*^ eyelids are defective in JNK phosphorylation, whether JNK is the downstream mediator in the control of eyelid closure remains an open question. A definitive answer to this question comes from genetic crossing experiments. Genetic crossing was used more than twenty years ago to study the pathways in relevance to the EOB phenotype. Specifically, two recessive mutants, Far and lgGa, were found to display non-allelic non-complementation in embryonic eyelid closure, such that when neither +/Far nor +/lgGa had eye defects, the double heterozygotes +/Far+/lgGa exhibited the EOB phenotype [63]. The authors surmised that Far and lgGa represented functionally related genes whose products worked in the same pathway [64]. The lgGa was mapped to Chr 13 and interestingly, it was later on shown to carry a deletion of the *Map3k1* gene [54].

The above studies pioneered a succession of genetic crossing experiments to search for players in the MAP3K1 network. The crossing of *Map3k1* and *Jnk1* mutants showed that neither *Map3k1*^Δ*KD/+*^ nor *Jnk1*^−/−^ mice had eye defects, but that their combinations, *Map3k1*^Δ*KD/+*^*Jnk1*^−/−^, resulted in the EOB phenotype [56]. Additionally, RhoA is a small GTPase implicated in MAP3K1 signaling in vitro [19,29,65]. RhoA conditional knockout in the ocular surface epithelium (*Rhoa*^Δ*OSE/*Δ*OSE*^) resulted in an eyelid closure delay by 2 days in the *Map3k1*^Δ*KD/+*^ background, but did not cause such a delay in the wild-type background [66]. The genetic data hence demonstrate the existence of a RhoA-MAP3K1-JNK pathway in embryonic eyelid closure.

The crossing experiments have further distinguished the differential roles of JNK1 and JNK2, two ubiquitously expressed mammalian JNK isoforms. JNK1 and JNK2 are known to have redundant developmental functions [67]. Mice lacking in either JNK1 or JNK2 are viable with no overt structural abnormalities, but double-deleted *Jnk1*^−/−^*Jnk2*^−/−^ mice die perinatally accompanied by multiple structural defects [68]. Crossing the *Jnk1* and *Jnk2* mutants with the *Map3k1*^Δ*KD*^ mice has led to a number of intriguing observations. While the *Map3k1* gene is haploinsufficient for embryonic eyelid closure in *Jnk1*-null mice, it is haplosufficient in *Jnk1*^+/−^ heterozygous mice, suggesting that the *Jnk1* allele contributes dose-dependently to MAP3K1 signaling. Additionally, unlike the *Map3k1*^Δ*KD*/+^*Jnk1*^−/−^ mice that display the EOB phenotype, the *Map3k1*^Δ*KD*/+^*Jnk2*^−/−^ mice have normal eye development [56]. However, the addition of a *Jnk2* mutation in the *Map3k1*^Δ*KD/+*^*Jnk1*^+/−^ background induces a partial EOB defect in the *Map3k1*^Δ*KD/+*^*Jnk1*^+/−^*Jnk2*^+/−^ mice, implying that JNK2 also makes contributions, albeit to a lesser extent, to the pathway. Altogether, these data show that while both JNK isoforms are involved in embryonic eyelid closure, JNK1 makes a greater contribution than JNK2 to MAP3K1 signaling, and that non-allelic non-complementation for the EOB phenotype can identify the molecular constituents of the MAP3K1 pathways in vivo.

### 3.3. Gene–Environment Interactions in Eyelid Morphogenesis

Besides the intrinsic genetic control, environmental factors can robustly modulate embryonic eyelid closure. Open-eye defects in mice were found following in utero exposure to pharmaceutical or environmental chemicals such as valproic acid [69], methamphetamine [70], anticonvulsants [71], lorazepam [72], methanol [73], and organophosphorus pesticides [74]. Additionally, hormones and retinoic acid treatment of pregnant mice prevented eyelid closure defects in the lgMl/lgMl genetic mutant mice, presenting a case of gene–environment interaction in eyelid morphogenesis. The identity of the lgMl mutant is still unknown, although it is evident that this gene function can be effectively modified by the maternal environment [75].

Another case of gene–environment interaction was found in the relationships between *Map3k1* and 2,3,7,8-Tetrachlorodibenzo-*p*-dioxin (TCDD). TCDD is an organochlorinated chemical and the prototype for a large number of environmental pollutants, collectively known as dioxin-like chemicals. These chemicals are global contaminants with wide-range toxicity, including a myriad of developmental toxicities [76,77,78]. In eyelid development, in utero TCDD exposure did not induce abnormalities in wild-type mice but caused defective eyelid closure in *Map3k1*^Δ*KD/+*^ fetuses, in parallel to a marked inhibition of phospho-JNK in the embryonic eyelid epithelium [79]. Most, if not all, TCDD toxicity is mediated by the aryl hydrocarbon receptor (AHR), a transcription factor encoded by the *Ahr* gene, activated by TCDD and many other dioxin-like pollutants [80]. In this context, the eyelid toxicity of TCDD was abolished in *Map3k1*^Δ*KD/+*^*Ahr*^−/−^ compound mutants, indicating that AHR mediated the crosstalk between MAP3K1 and TCDD.

EGFR and DKK2, two members of the EGFR and WNT pathways, respectively, are also crucial for embryonic eyelid closure. Similar to *Map3k1*, both *Egfr* and *Dkk2* homozygous deletions, but not the hemizygous, induce the EOB defects [81,82]. However, different from *Map3k1*, the hemizygous *Egfr*^+/−^ and *Dkk2*^+/−^ do not potentiate the toxicity of TCDD in eyelid closure, demonstrating a remarkable specificity of the pathways that TCDD-AHR interacts with [79]. The existence of the MAP3K1-TCDD signaling crosstalk raises an intriguing possibility that other MAP3K1 network genes, such as *Jnk* and *RhoA*, can also modulate the developmental toxicity of TCDD.

### 3.4. MAP3K1 Signaling Is a Developmental Threshold

Based on the genetic crossing data, Harris and Juriloff postulated that eyelid morphogenesis depends on a developmental threshold [63]. The molecular identity of the threshold had been elusive for the past twenty years, until MAP3K1 came into the picture. MAP3K1 signaling is a tangible threshold to which the different genes, i.e., *Map3k1*, *Jnk1*, *Jnk2,* and *RhoA*, contribute with a gradient of strength (Figure 3A). Of these genes, *Map3k1* makes the strongest contribution as its homozygous mutants produce the EOB phenotype; *Jnk1* is weaker because the *Jnk1*^−/−^ deletion requires *Map3k1* hemizygosity to produce the defects; *RhoA* and *Jnk2* are still the weakest since they require second or third gene mutations and induce milder symptoms. The genetic mutations cumulatively inhibit MAP3K1 signaling, which, when decreased to below the threshold levels, triggers eyelid closure failure. The TCDD-induced eyelid defects in *Map3k1*^Δ*KD/+*^ mice also fit nicely in the threshold model, in which the eye defects occur when the combined genetic and environmental insults cause a significant inhibition of MAP3K1 signaling.

### 3.5. Complex Genetic Control of Embryonic Eyelid Closure

Deficiency of embryonic eyelid closure does not abrogate embryonic and fetal survival, but results in an EOB phenotype that is extremely easy to recognize in the newborn pups [83]. For this reason, more than 140 mutant strains in the Mouse Genome Informatics (MGI) database are reported to have the EOB phenotype in addition to the *Map3k1* mutants. An investigation of the mutants has led to the findings that eyelid closure requires: (i) the RA-RXR/RAR and PITX2-DKK2 pathways; (ii) the FOXL2 and OSR2 transcription factors operating in the periocular mesenchyme; and (iii) the activation of the MAP3K1, EGFR, ROCK, and PCP pathways in the eyelid epithelial cells [84,85,86,87,88,89,90,91,92]. The FGF10-FGFR and BMP-BMPR pathways, on the other hand, regulate eyelid morphogenesis by acting through mesenchymal–epithelium interactions [93,94]. Some of the signaling events are detected in just a few cells, where morphogenesis is regulated by compartmentalized and spatially segregated local signals [61,95,96]. The molecular mechanisms underlying embryonic eyelid closure may serve as a good model for understanding other genetically complex morphological processes in mammals.

### 3.6. Congenital Eye Defects Associated with the EOB Phenotype

The large number of genetic mutants displaying the EOB defects suggests that there are many situations in which gene mutations will induce eyelid closure failure and the subsequently congenital eye defects. Analyses of *Map3k1*^Δ*KD/*Δ*KD*^ and other EOB mice have identified numerous types of corneal pathogenesis, including epithelial erosion, hyperplasia, squamous metaplasia, and corneal stroma neovascularization [53,97]. As the mouse cornea is immature at birth and becomes fully developed on postnatal day (P) 12—timing that coincides with eye opening [98]—the closed eyelids are speculated to offer protection to the immature corneas from environmentally induced injuries.

The EOB mice also display multiple developmental abnormalities of the ocular adnexa [99]. A number of ocular adnexal tissues become fully developed while the eyelids are closed. When the eyelids are fused, the *orbicularis oculi* muscle and tarsal plate become mature, the smooth muscles attach to the tarsal plate responsible for eyelid elevation, and meibomian glands start to form to produce meibum that prevents evaporation of the tear film [100,101,102]. Without eyelid closure, the tarsal muscles are truncated, *inferior oblique* muscles are misplaced in the prenatal E18.5 fetuses, and eyelid tarsal muscles are hypoplastic, *LPS* extension is blunted, and *inferior rectus* muscles are mis-located in the adult mice [99,103]. The EOB mice also have hypoplastic and ill-formed meibomian glands [104]. Taken together, the analyses of the EOB mice suggest that eyelid closure deficiency may lead to congenital corneal dystrophy and adnexal abnormalities.

## 4. MAP3K1 Regulates Sexual Determination and Differentiation

Mammalian sex organ development begins at the embryonic stages, where two sets of ducts, i.e., the Müllerian ducts (MDs) and the Wolffian ducts (WDs), arise from the mesonephric kidneys [105]. The ducts form in both sexes in early embryogenesis, and after sex determination, the testis-derived hormones in males promote WD elongation while triggering MD regression. The male WDs continue to develop into separate but contiguous organs, including epididymis, *vas deferens*, and seminal vesicles. Conversely, owing to the lack of testis-derived hormones in females, the WDs regress, and the MDs elongate and differentiate. The MDs further develop into the female reproductive tracts composed of the fallopian tubes, uterus, cervix, and vagina. Many genes and signaling pathways play crucial roles in sex and reproductive tract development; their deficiency leads to anomalies in sex determination and differentiation [106].

The 46, XY disorder of sex development (DSD) is a condition in which a genetically male individual has ambiguous or feminized genitalia, complete or partial gonadal dysgenesis, abnormal testis, and reduced to no sperm production [107]. Accumulated clinical genomics data show that approximately 13–18% of 46, XY DSD patients carry *MAP3K1* allelic variants [106,108,109,110,111,112,113,114,115]. The pathogenic mutations are found throughout the *MAP3K1* gene, but the majority are non-synonymous changes affecting the GEF and ARM domains of the protein. One of the first mutations identified in familial cases was a splice-acceptor mutation (634-8T > A) that results in the addition of two amino acid residues, leucine and glutamine, in the GEF domain [114]. Missense mutations of L189 in the GEF domain and P153 substitution of Leucine at the N-terminal to the GEF domain were subsequently found in sporadic 46, XY DSD patients [111,113]. Mutations affecting the ARM domain of MAP3K1, including the loss of 34 amino acid residues (G727-I761del) and missense substitutions, G616R and L706R, were also identified in both familial and sporadic cases. Additionally, a genetic variant resulting in an A1433V mutation in the kinase domain was reported in two sporadic 46, XY DSD patients [106]. Up to date, *MAP3K1* is considered one of the most commonly mutated genes associated with 46, XY DSD.

The genetic variants affecting the GEF and ARM domains exhibit gain-of-function properties in signal transduction. In vitro studies showed that the mutated MAP3K1 displayed enhanced binding with co-factors, i.e., RHOA and MAP3K4, and caused the increased activation of the downstream targets such as p38 and ERK1/2 [17,113,114,115,116]. Hyperactivation of p38 and ERKs might in turn activate the WNT-β-catenin pathway to block SOX9 function, which is required for male sex development. Thus, mutations of MAP3K1 tilt the balance of sex development and differentiation through the dysregulation of WNT signaling.

## 5. Conclusions and Perspectives

Studies of mouse models offer significant insights into MAP3K1 signaling in eyelid morphogenesis. First, MAP3K1 orchestrates a signaling network—the different network genes display non-allelic non-complementation in embryonic eyelid closure. Besides *Map3k1*, *Jnk1*, *Jnk2,* and *Rhoa*, other components of the network are yet to be identified. For example, Far, mapped to Chr. 2, exhibits non-allelic non-complementation with *Map3k1*^+/−^ for an EOB phenotype [63], suggesting that Far is a component of the same network [54,63]. The genetic identity of Far, up to the present date, remains unknown. Second, MAP3K1 signaling is a developmental threshold for eyelid morphogenesis, leading to the proposition that any genetic or environmental insults, or their combinations, that repress MAP3K1 signaling could increase the risks of and possibly cause defects in the eye (Figure 3A). The threshold model may serve as a theoretical basis for predicting the polygenic and multifactorial etiology of eyelid morphogenetic disorders due to the complex interactions of many genetic variants and environmental chemicals.

Translation of the knowledge from mouse studies in order to understand human eye development and diseases is lacking. The developmental timeline of eyelids in human and mice is similar [100,101,117,118,119], but unlike mice, the closure and re-opening of the human eyelid is accomplished entirely in utero, occurring between 7 and 24 weeks of fetal life. This has made it challenging to detect lid closure defects in humans. Up to date, little is known about the human incidence of eyelid closure defects and the associated disease phenotypes. If the developmental eyelid closure serves analogous functions in mice and humans, its deficiency in humans will likely increase the risks of maternal exposure-induced ocular surface anomalies, such as corneal clouding and dystrophies, and adnexal abnormalities, such as congenital blepharoptosis and strabismus. These human disorders are not uncommon; their pathological origins are diverse and still poorly understood.

The major breakthrough in the translational front is the compelling human genomics data associating *MAP3K1* variants with the 46, XY DSD syndrome. Mechanistically, the mutant MAP3K1 has gain-of-function properties, ultimately leading to the hyperactivation of WNT that in turn crucially regulates sex determination and differentiation [17,105,120]. This mechanism, however, has not been validated in vivo in an animal model because a MAP3K1 gain-of-function mouse, to our knowledge, has not been developed.

In mice, MAP3K1 expression is detected in the developing reproductive tissues of both sexes [114,121]. If MAP3K1 gain-of-function up-regulates WNT, its loss-of-function accordingly may lead to WNT hypoactivation, which is linked to abnormalities in female sex differentiation (Figure 3B). The ablation of WNT isoforms in mice indeed results in diverse female reproductive disorders [122,123,124,125]; in humans, WNT4 missense mutations are associated with 46, XX DSD and MD anomaly [126,127,128]. The *Map3k1*-null mice, without major defects in testis formation, have not been carefully examined for female sex tissues [121]. Investigating *Map3k1* loss-of-function in female sex development in the mouse models will likely shed new light on and consolidate the MAP3K1–WNT axis in sex determination and differentiation.

## Figures and Tables

**Figure 1 cells-11-00034-f001:**
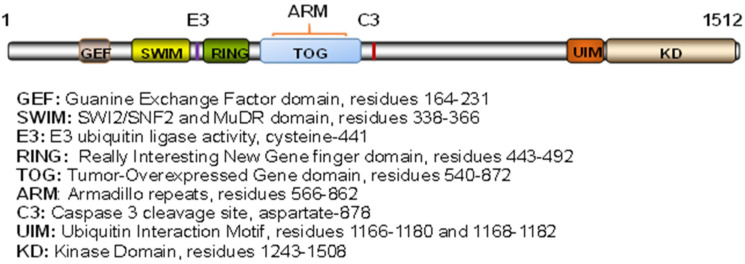
Schematic structure of MAP3K1. Illustration of the functional domains and the relative domain locations of human MAP3K1.

**Figure 2 cells-11-00034-f002:**
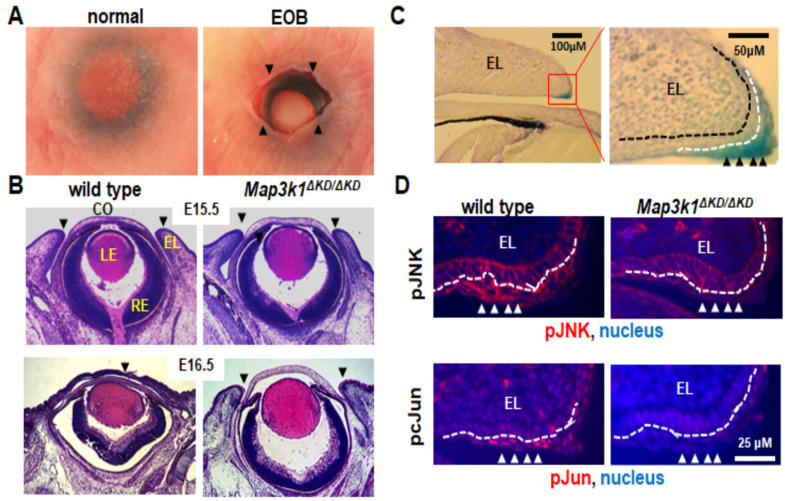
Mouse embryonic eyelid development and closure. (**A**) Photographs of eyes of the newborn pups. Mice are normally born with eyelids closed, but those with defective embryonic eyelid closures are born with an EOB phenotype. Arrowheads point at the eyelid opening margins in EOB mice. (**B**) Histological analyses of the embryonic eyes at E15.5 and E16.5. At E15.5, the wild-type and *Map3k1*^Δ*KD/*Δ*KD*^ embryos have the same eye structures, with the upper and lower eyelids separated. At E16.5, the eyelids were fused in the wild-type, but remained separated in the *Map3k1*^Δ*KD/*Δ*KD*^ fetuses. Arrowheads point at the leading edge or fusion junction of the upper and lower eyelids. (**C**) Eye sections of the whole-mount X-gal stained *Map3k1^+/^*^Δ*KD*^ E15.5 embryos were photographed at 10× (left panel), the red box in the left panel was shown at 40× magnifications (right panel). Abundant β-gal positive, i.e., MAP3K1-expressing cells were detected in the eyelid epithelium, particularly in the inferior periderm near the eyelid tip. (**D**) Immunohistochemistry analyses with antibodies for the pJNK (upper panels) and p-cJun (lower panels) of the E15.5 eyes. The pJNK and pJun were detected in the inferior periderm near the eyelid tip in wild-type but not *Map3k1*^Δ*KD/*Δ*KD*^ embryos. Nuclei were stained with DAPI (blue). EL: eyelid, CO: cornea, LE: lens, RE: retina. Black dotted lines mark the basement membrane; white dotted lines mark the boundary between the basal epithelium and periderm. Arrowheads point at the MAP3K1-expressing periderm.

**Figure 3 cells-11-00034-f003:**
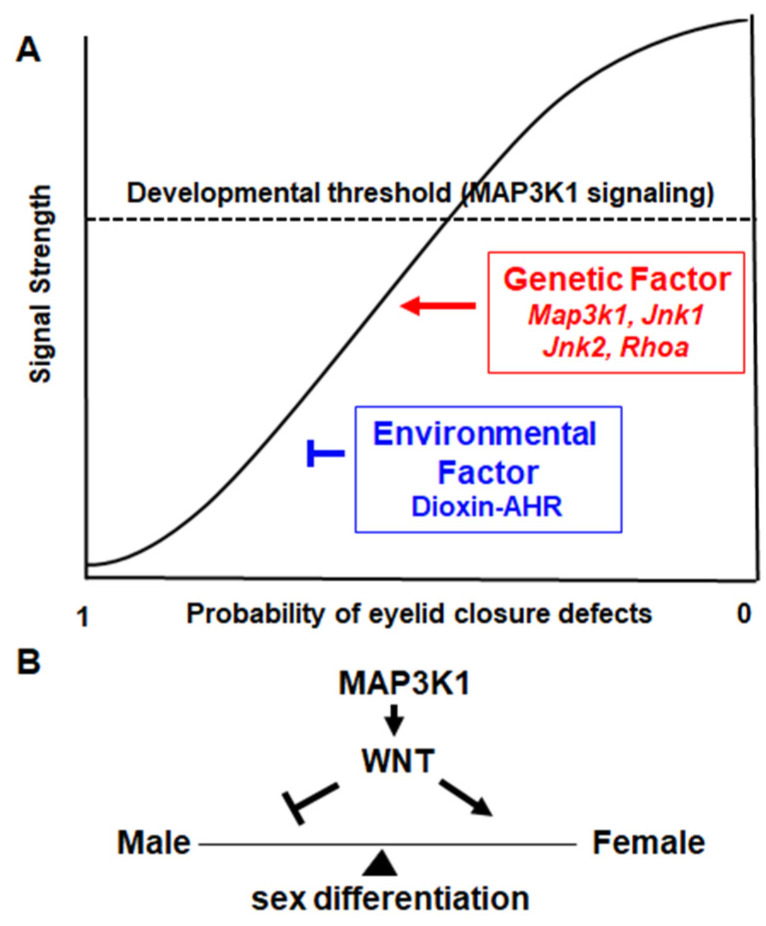
Summary and perspectives. (**A**) MAP3K1 signaling serves as a developmental threshold. Different genes and environmental factors distinctly affect the strength of MAP3K1 signaling; when the signaling strength reduces to levels below the critical threshold eyelid closure abnormalities occur. (**B**). The hypothetical roles of the MAP3K1–WNT axis in maintaining the balance of male–female sex differentiation.

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
