# Peer review of "Genetic Control of MAP3K1 in Eye Development and Sex Differentiation"

_cells, 2021, doi:10.3390/cells11010034_

Round 1

Reviewer 1 Report

In this work the authors analysed the role of MAP3K1 in regulating biological functions. Generating mice with ablation of the entire gene, the kinase domain coding sequences, or harboring point mutations in the MAP3K1 gene, the authors were able to determine the diverse roles that MAP3K1 plays in embryonic survival, maturation of T/B cells, and development of sensory organs, including eye and ear. In addition, detailed analyses focused on eye development have further elucidated a possible molecular network in which MAP3K1 integrates genetic and environmental signals.

Overall, the review is clear and the importance of MAP3K1 is described in great details.

This work may be considered for publication in Cells

Author Response

We thank the reviewer for the time invested in reading the paper.  No specific responses are required for this review.

Reviewer 2 Report

The review article entitled "Genetic control of MAP3K1 in development and disease" by J. Wang et al. Discusses different transgenic mice models to address the biological functions of the MAP3 kinase MEKK1 mainly in terms of the development of sensory organs, especially the eye. There are several topics that I think should be added by the authors.

  1. In the abstract,the authors are enumarate different biological activities of MEKK1, including embryonic survival, maturation of B and T cells, development of the eye and the ear, sex development and differentiation. In fact, the authors discuss mainly the development of ther eye. In this regard is the title of the article misleading, because the reader expect a comprehensive descriptionof MEKK1 “in development and disease”. The authors should either expand the article to discuss the mentioned topics in more detail, or change the title of the article.

  1. The article does not give any information about the activation procedure of MEKK1, which is essential for a stimulus-induced protein kinase. Which extracellular signals activate MEKK1 and what is the mechanism of the activation.

  1. I am missing the discussion of the protein-protein interactions of MEKK1 (Rho, Rat, Cdc42, TRAF2, Grb2, Axin, alpha-actinin, NIK etc.) and their functions in controlling the biological activity of MEKK1.

  1. There is also no discussion about the role of MEKK1 (auto)phosphorylation and dephosphorylation by protein phosphatases ? What is the role of auto-ubiquitylation of MEKK1 ?

  1. Discussing a protein kinase it es essential to know the substrates. Which substrates are known for MEKK1? What does the substrates tell us about the biological function ?

  1. Controversal issues may also be discussed in the article. For example, what is the relationship between the activation of the JNK/c-Jun pathway by MEKK1 and the binding and ubiquitylation of c-Jun by MEKK1. Likewise, what is the activation potential of the ERK1/2 pathway by MEKK1?

Author Response

We thank the reviewer for the comments. Following please find our responses to specific critiques of the reviewer:

1. In the abstract, the authors are enumarate different biological activities of MEKK1, including embryonic survival, maturation of B and T cells, development of the eye and the ear, sex development and differentiation. In fact, the authors discuss mainly the development of the eye. In this regard is the title of the article misleading, because the reader expect a comprehensive description of MEKK1 “in development and disease”. The authors should either expand the article to discuss the mentioned topics in more detail, or change the title of the article.

Response: The title is modified as “Genetic Control of MAP3K1 in Eye Development and Sex Differentiation” to better reflect the main contents of the paper.

2. The article does not give any information about the activation procedure of MEKK1, which is essential for a stimulus-induced protein kinase. Which extracellular signals activate MEKK1 and what is the mechanism of the activation. I am missing the discussion of the protein-protein interactions of MEKK1 (Rho, Rat, Cdc42, TRAF2, Grb2, Axin, alpha-actinin, NIK etc.) and their functions in controlling the biological activity of MEKK1. There is also no discussion about the role of MEKK1 (auto)phosphorylation and dephosphorylation by protein phosphatases? What is the role of auto-ubiquitylation of MEKK1? Discussing a protein kinase it es essential to know the substrates. Which substrates are known for MEKK1? What does the substrates tell us about the biological function?  Controversal issues may also be discussed in the article. For example, what is the relationship between the activation of the JNK/c-Jun pathway by MEKK1 and the binding and ubiquitylation of c-Jun by MEKK1. Likewise, what is the activation potential of the ERK1/2 pathway by MEKK1?

Response: This article is not intended to cover all molecular biological aspects of MAP3K1. Rather, its focuses are on understanding how MAP3K1 operates in vivo in developmental models. We hope this point is clarified now with the revised Title and Abstract. Nevertheless, we addressed some of the reviewer’s comments and included descriptions on 1) signals that activate MAP3K1, 2) the MAP4Ks that interact and phosphorylate MAP3K1, 3) ubiquitination substrates, and 4) MAP3K1 autophosphorylation and glutathionylation, noting that the descriptions are kept brief to minimize distractions from the main focus.

Reviewer 3 Report

The important role of mitogen-activated protein kinases (MAPKs) in diverse cellular pathways cannot be negated. There are many studies and review articles published discussing their role in both development and diseases both infectious and non-infectious in nature (Lawrence, M. C., et al,  Cell Research (2008) 18:436-442; Kim, E. K. & Choi, E., Biochimica et Biophysica Acta 1802 (2010) 396–405; Mohanta, T. K., et al, Biomed Res Int., 2020 Dec 31;2020:8827752; Chen, J., et al, Int. J. Mol. Sci. 2021, 22, 9640). The purpose of a review article is to provide most updated information on a topic. Though the topic of this review article is interesting, the most of article articulates around eye-open at birth (EOB) phenotype. Moreover, there is no significant description on the role of MAP3K1 in diseases.

  1. There are potential issues in the review article. AT some places, it seems that author did research and are sharing their findings.
  2. Sections 2 and 3 have redundant information. The purpose of the review article is to streamline the information so that reader can have a clear view about the information. After reading both sections, it seems that authors have collected information from various papers.
  3. The introduction part should have clearly discussed the cascade of activation between MAPKKK, MAPKK and MAPK. Furthermore, they should have used conventional terms for MAPK such as MAP3K1 is also known as MEKK1 etc.
  4. Line 202: Is c-Jun expression or phosphorylation affected?
  5. Line 57-59: Armadillo repeats (ARM) should be shown in figure 1.
  6. Line 79: ref. needed
  7. Figure 2: what is the source of this figure? If taken from a published article, permission was taken?
  8. Lines 205-207: Give some information on spatiotemporal activation of c-Jun and MAP3K1 activation/induction in embryo.
  9. Lines 221-224: the authors have mentioned that there was reduction in c Jun levels due to Map3k L34Q/+ mutation. Did the authors of that manuscript effect on phosphorylation of c Jun? It seems from the sentence that this mutation affected transcription/translation levels of c Jun.

Author Response

We thank the reviewer for carefully reading and commenting on the paper. Below please find our responses to the specific critiques:

1. There are many studies and review articles published discussing their role in both development and diseases both infectious and non-infectious in nature (Lawrence, M. C., et al,  Cell Research (2008) 18:436-442; Kim, E. K. & Choi, E., Biochimica et Biophysica Acta 1802 (2010) 396–405; Mohanta, T. K., et al, Biomed Res Int., 2020 Dec 31;2020:8827752; Chen, J., et al, Int. J. Mol. Sci. 2021, 22, 9640). The purpose of a review article is to provide most updated information on a topic. Though the topic of this review article is interesting, the most of article articulates around eye-open at birth (EOB) phenotype. Moreover, there is no significant description on the role of MAP3K1 in diseases. Response: The reviewer is absolutely correct in that this paper focuses on topics of MAP3K1 that are new and/or not extensively covered elsewhere. There might be confusions, stemmed in part from the vagueness of the Title and the Abstract, which are now revised in the resubmission.

2. Sections 2 and 3 have redundant information. The purpose of the review article is to streamline the information so that reader can have a clear view about the information. After reading both sections, it seems that authors have collected information from various papers. Response: Sections 2 and 3 are reorganized and re-written to minimize redundancy.

3. The introduction part should have clearly discussed the cascade of activation between MAPKKK, MAPKK and MAPK. Response: Activation cascades through phosphorylation are added.

4. Furthermore, they should have used conventional terms for MAPK such as MAP3K1 is also known as MEKK1 etc. Response: This information was included in the paper, even though we strive to use standard nomenclature when possible.

5. Line 202: Is c-Jun expression or phosphorylation affected? Response:  This point is clarified in the revision.

6. Line 57-59: Armadillo repeats (ARM) should be shown in figure 1. Response:  added

7. Figure 2: what is the source of this figure? If taken from a published article, permission was taken?  Response: To avoid complications, we exchange all images with a set of unpublished data.

8. Lines 205-207: Give some information on spatiotemporal activation of c-Jun and MAP3K1 activation/induction in embryo. Response: The information is added in the revision.  

9. Lines 221-224: the authors have mentioned that there was reduction in c Jun levels due to Map3k L34Q/+ mutation. Did the authors of that manuscript effect on phosphorylation of c Jun? It seems from the sentence that this mutation affected transcription/translation levels of c Jun. Response: Description is re-written to clarify that the Map3k L34Q/+ mutant has decreased c-Jun expression and phosphorylation.

Round 2

Reviewer 2 Report

The authors habe changed the title of the article according to my suggestions. In addition, they provide more details about the biochemistry of MEKK1. Thus, I am recommending to accede the manuscript.

Some minor remarks:

Page 2, line 23: developmetnal

Figs. 1 and 2 are shown twice, either with or without a legend.